# Carbon Quantum Dots-Functionalized UiO-66-NH_2_ Enabling Efficient Infrared Light Conversion of 5-Hydroxymethylfurfuryl with Waste Ethanol into 5-Ethoxymethylfurfural

**DOI:** 10.3390/ijerph191610437

**Published:** 2022-08-22

**Authors:** Hong Xiao, Yunting Zhang, Junran Gong, Kexin Li, Xing Chen, Dexin Fang, Guochun Lv, Ganxue Wu, Shihuai Deng, Zhenxing Zeng

**Affiliations:** 1College of Environmental Sciences, Sichuan Agricultural University, Chengdu 611130, China; 2ZHTH Research Institute of Environmental Sciences, Beijing 100085, China; 3Key Laboratory of Jiangxi Province for Persistent Pollutants Control and Resources Recycle, Nanchang Hangkong University, Nanchang 330063, China

**Keywords:** acid catalysis, 5-methoxymethylfurfural synthesis, immobilized catalyst, biological carbon quantum dots, photothermal conversion

## Abstract

The catalytic etherification of 5-hydroxymethylfurfural (HMF) with the waste ethanol into high-energy-density 5-ethoxymethylfurfural (EMF) has been considered as a promising way to simultaneously alleviate the energy crisis and environmental pollution. However, the energy consumption is rather high as the synthesis of EMF requires a high temperature to open the etherification reaction. Herein, we demonstrate a clever design and construction of acidified biomass-derived carbon quantum dots (BCQDs)-modified UiO-66-NH_2_ that is immobilized on cermasite (H^+^/BCQDs/UiO-66-NH_2_@ceramsite), which can use the IR light as driven energy and wasted ethanol to trigger the catalytic conversion of HMF into EMF. The temperature on the surface of the immobilized catalyst could reach as high as 139 °C within 15 min IR irradiation. Due to the aforementioned advantages, the as-prepared catalyst exhibited excellent IR-triggered catalytic performance toward EMF production, where the EMF yields and selectivity were as high as 45% and 65%, respectively. The high catalytic performance originates from the outstanding photo-to-thermal conversion by the introduction of BCQDs, as well as the strong interactions between BCQDs and UiO-66-NH_2_ that boosts the etherification reactions. The immobilization of catalyst on cermasite not only benefits catalyst recycling, but more importantly reduces catalyst loss during practical applications. The conceptual study shown here provides new viewpoints in designing energy-effective materials for the conversion of wastes into high-value-added resources.

## 1. Introduction

Energy shortage and environmental pollution caused by fossil fuel burning in recent years have been regarded as two major problems that affect our daily life and social developments [1,2]. In this regard, the exploration of clean fuels, for example, renewable biofuels, as an alternative to the traditional fossil fuel has been shown as a promising way to overcome the aforementioned problems [3,4]. Among various biofuels, 5-ethoxymethylfurfural (EMF) with the energy density of 30.3 MJ·L^−1^ (comparable to that of diesel (33.6 MJ·L^−1^) and gasoline (30.6 MJ·L^−1^)) has been considered as a promising alternative to traditional fossil fuels [5,6,7,8,9,10]. Furthermore, the use of EMF instead of fossils could also decrease soot and sulfur oxide emissions, which may have a positive effect in alleviating the air pollution [11,12,13]. Generally, EMF is synthesized through the classical reaction of etherification of 5-hydroxymethylfurfural (HMF) through an acidic catalysis manner [14,15,16,17,18]. Since the catalyst is the core part of the catalytic reaction, the key to achieve high HMF synthetic performance lies in the exploration of highly efficient acid catalysts. 

Various acid catalysts, such as sulfonated carbon [19], zeolites [20,21], sulfated zirconia [22,23], ion-exchange resins [24], and HNbMoO_6_ [8], have been explored as acid catalysts for the conversion of HMF into EMF. Although considerable progress has been made during the past decades, we notice that almost all the developed acid catalysts are capable of etherifying the HMF into EMF only under a high temperature (mostly higher than 100 °C) [25]. Traditionally, maintaining such a high temperature of the catalytic system during the synthetic procedure has been energy-consuming, often involving the combustion of other kinds of energy. This is unacceptable as the production of EMF is at the cost of other kinds of energy, which may largely increase the cost for EMF synthesis and restrict the practical applications. Thus, the development of a novel self-powered acid catalyst capable of converting HMF into EMF is urgently needed. What about using photothermal material as an acid catalyst? The use of photothermal material as an acid catalyst could convert the intermittent solar energy into heat energy to warm up the catalytic system, speeding up the conversion of HME into EMF. However, most photothermal materials cannot be directly used as acid catalysts to drive the etherifying reaction due to the lack of active sites. In this regard, modifying the photothermal material with functional groups to endow it with catalytic activity is therefore urgently needed. To the best of our knowledge, the establishment of self-powered acid catalytic system for the conversion of HMF into EMF has not been reported and is yet to be explored. 

Among various photothermal materials, carbon quantum dots (CQDs) with an outstanding solar-to-heat conversion property and an adjustable surface structure are a potential material for acid catalysis [26,27,28]. On the one hand, the abundant oxygen-containing groups, including hydroxyl group on the surface of CQDs, make it easy to functionalize with catalytic species to endow it with acid catalytic activity [29]. On the other hand, the excellent solar-to-heat conversion property could warm up the local temperature to open the catalytic reaction of converting HMF to EMF. Modifying the CQDs with an amino-functionalized Zr-based metal–organic framework (UiO-66-NH_2_) via amino hydroxylation is a better choice to endow the composite with catalytic activity, as the amino groups on UiO-66 can be acidified [30,31]. Nevertheless, the use of powder-form catalyst suffers from catalyst separation. Thus, it is still needed to immobilize the catalyst so as to overcome the aforementioned drawbacks. 

In this work, we demonstrate a clever design of acidified biological carbon quantum dots (BCQDs)-modified UiO-66-NH_2_ (H^+^/BCQDs/UiO-66-NH_2_) composite and the immobilization of it on ceramsite (H^+^/BCQDs/UiO-66-NH_2_@Ceramsite) for the photothermal synthesis of EMF. The abundant -NH_2_ groups on UiO-66 not only improve the adhesion between BCQDs and the ceramsite, but more importantly promote the linkage of H^+^ to serve as active sites. Furthermore, the black-colored BCQDs serving as photo-to-thermal convertor could provide a high-temperature environment to open the etherification reaction. The temperature on the surface of the immobilized catalyst could reach as high as 139 °C within 15 min IR irradiation. Benefited from the aforementioned advantages, the as-prepared catalyst exhibits an excellent IR-triggered catalytic performance toward EMF production, where the EMF yields and selectivity were shown to be as high as 45% and 65%, respectively. As a result of the aforementioned advantages, the as-prepared H^+^/BCQDs/UiO-66-NH_2_ exhibits excellent infrared-light-triggered photothermal conversion of HMF for EMF synthesis. The innovative study shown in this work may provide new viewpoints in designing a highly efficient acid catalytic system for EMF production.

## 2. Experimental

### 2.1. Chemicals and Reagents

Waste *Camellia oleifera* shells (abbreviated WCOSs) were received from Jiangxi Green Sea Oil Co., Ltd., Ji’An, China. The as-received WCOSs were subjected to dehydration, crush, and sieve treatment before further use. Zirconium chloride (ZrCl_4_ ≥ 99.5%), 2-aminoterephtalic acid (BDC-NH_2_, ≥99%), phloroglucinol (C_6_H_6_O_3_, ≥99.0%), and 5-hydroxymethylfurfural (C_6_H_6_O_3_, 98%, abbreviated HMF) were purchased from Shanghai Macklin Biochemical Co., Ltd., Shanghai, China. N, N-dimethylformamide (DMF, ≥99.5%) was purchased from Guangzhou Xilong Chemical Co., Ltd., Guangzhou, China. 5-Ethoxymethylfurfural (C_8_H_10_O_3_, 97%, abbreviated EMF) was purchased from Sigma-Aldrich (Shanghai) Trading Co., Ltd., Shanghai, China. Ceramsite (n represents the particle size of ceramsite; n = 4~6 mm) was purchased from Huaxiangu agriculture Co., Ltd., Xiamen, China. All chemicals except that of waste *Camellia oleifera* shells were used as-received without further purification. 

### 2.2. Materials Preparation

The UiO-66-NH_2_ was fabricated via a facile hydrothermal synthetic procedure. Typically, 90 mg of ZrCl_4_ was dissolved in 35 mL of DMF under vigorous stirring until the solution become transparent; afterwards, another mixture (prepared by dissolving 60 mg of BDC-NH_2_ in 35 mL of DMF) was added into the above solution under ultrasonication treatment to form a transparent light-yellow mixture. After stirring for another 10 min, the as-obtained mixture was subjected to hydrothermal treatment in a 100 mL Teflon at 160 °C for 24 h to allow the formation of MOF material. After the hydrothermal treatment, the Teflon was cooled down to room temperature naturally and the precipitates were collected by centrifugation. The collected pale-yellow precipitates were further washed with ethanol 3 to 4 times, and after drying in an oven at 100 °C for 12 h, the UiO-66-NH_2_ was obtained.

Bio-based carbon quantum dots (BCQDs) suspension with a certain concentration was prepared directly from WCOSs according to our previously reported method [32]. For the synthesis of BCQDs/UiO-66-NH_2_, 200 mg of UiO-66-NH_2_ was dispersed into 40 mL of as-prepared BCQDs solution under ultrasonication treatment for about 10 min. After stirring the obtained suspension at 80 °C for 4 h, the mixture was centrifuged and dried in a vacuum oven overnight. To strengthen the interactions between the BCQDs and UiO-66-NH_2_, the composite was calcinated at 200 °C for 3 h. 

For the synthesis of H^+^/BCQDs/UiO-66-NH_2_, 100 mg of BCQDs/UiO-66-NH_2_ was dispersed in 50 mL of dilute hydrochloric acid and stirred under room temperature for 3 h. Subsequently, the suspension was centrifuged and washed with deionized water until the solution reached neutral, and the collected precipitate was vacuum-dried overnight at 60 °C. As a reference, H^+^/UiO-66-NH_2_ was prepared by the direct acidification of UiO-66-NH_2_ in dilute hydrochloric acid using the same method. 

To make it easy for catalyst recycle, the as-prepared powder-form catalyst was immobilized on lightweight ceramsite. In a typical procedure, 100 mg of UiO-66-NH_2_ and 2.8 g of ceramic particles were added into 5 mL of ethanol, and the mixture was then subjected to ultrasonication treatment to allow the deposit of MOF onto the ceramsites. After drying ceramic particles at 100 °C for 12 h in a vacuum oven, the ceramic particles were further subjected to thermal treatment in a tube furnace for 2 h in Ar atmosphere under a certain temperature (250, 300, 400, 500 °C), and the resulting particles were collected and donated as UiO-66-NH_2_@Ceramsite. To fabricate the BCQDs/UiO-66-NH_2_@Ceramsite, the as-prepared UiO-66-NH_2_@Ceramsite was then immersed into 40 mL of BCQDs solution and heated at 80 °C for 4 h under vigorous stirring. After separation, the functionalized ceramsite was dried and calcinated at 200 °C for another three hours. The final brown-colored material is referred to BCQDs/UiO-66-NH_2_@Ceramsite. To install the material with catalytic sites, the as-obtained BCQDs/UiO-66-NH_2_@Ceramsite was allowed to immerse into 50 mL of dilute hydrochloric acid, and the mixture was kept at room temperature for 3 h. After washing and drying the ceramic particles, the desired H^+^/BCQDs/UiO-66-NH_2_@Ceramsite was then collected. 

### 2.3. Characterizations

The microstructure was studied by using transmission electron microscopy (TEM), high-resolution transmission electron microscopy (HRTEM), and field-emission scanning electron microscopy (FESEM). The as-prepared acid catalysts were further tested by using a Samrtlab (9) Powder X-ray diffraction (XRD) diffractometer equipped with a Cu-Kα radiation to gain the crystalline structure information. As the porosity of the catalyst could largely affect the mass transfer process during the catalytic reaction, the surface area and porosity of the catalyst were studied using a Nova-2000e machine. The chemical states of as-prepared catalysts were tested on a monochromated Al-Kα X-ray photoelectron spectroscope (XPS, Axis Ultra DLD) at a residual gas pressure of less than 10^−8^ Pa. All the binding energies were referenced to the C 1s peak at 285 eV of the surface adventitious carbon. The surface groups of catalyst were further investigated by using a NicoletiS5 FTIR apparatus. As the light adsorption property of the material is an important factor affecting the photothermal conversion, the as-prepared materials were further investigated by using a Lambda 750S UV/VIS/NIR spectrometer.

### 2.4. Catalytic Activity Evaluation

The catalytic reaction was carried out in a homemade reaction; during the catalytic reaction the reaction system was continuously bubbled with nitrogen gas to guarantee the anaerobic condition. In detail, 50 mg of powder-form catalyst was added into the flask, then 63 mg of HMF and 25 mL of ethanol were added into the above flask to form a uniform suspension. The reaction system was heated to 100 °C and held at this temperature for 24 h to allow the etherification of HMF with EtOH for EMF production. After the reaction, 0.1 mL of the reaction solution was sampled out and diluted with acetonitrile and ultrapure water. The concentrations of EMF and HMF were analyzed by using an Agilent 1100 series high-performance liquid chromatography (HPLC) system equipped with a C18 column and DAD detector. 

An infrared lamp (150 W, Philips Co., Ltd., Warszawa, Poland), simulating the solar light, was employed as input driven energy to trigger the etherification reaction. The output wavelength was set as λ > 750 nm. Typically, 50 mg of H^+^/BCQDs/UiO-66-NH_2_ or 2.8 g of H^+^/BCQDs/UiO-66-NH_2_@Ceramsite (containing about 50 mg of catalyst) was dispersed in solution containing 63 mg of HMF and 25 mL of ethanol. The reaction system was bubbled with nitrogen gas and irradiated with IR light to allow the etherification of HME. The catalytic performance of as-prepared catalysts was evaluated by using the factors of EMF yield and generation selectivity, and HMF conversion rate. Here, the EMF yield, HMF conversion rate, and EMF selectivity were calculated by using the formulae of EMF yield (%) = (N_B_/N_A0_) × 100 and EMF selectivity (%) = [N_B_/(N_A0_ − N_A_)] × 100, respectively. Among them, N_A0_ is the initial molar amount of HMF, N_A_ is the residual molar amount of HMF after the reaction, and N_B_ is the molar amount of EMF after the reaction.

## 3. Results and Discussion

### 3.1. Materials Characterizations

The powder-form acid catalyst of H^+^/BCQDs/UiO-66-NH_2_ is synthesized via a combined synthetic strategy of immerse and acidification treatment, where the as-prepared UiO-66-NH_2_ is firstly immersed in the BCQDs solution to obtain BCQDs/UiO-66-NH_2_, then the composites are further acidified by hydrochloride acid to form H^+^/BCQDs/UiO-66-NH_2_. As shown in Figure 1, H^+^/BCQDs/UiO-66-NH_2_ features a particle morphology with a dense, stacked structure. An enlarged observation indicates the amorphous structure of the UiO-66-NH_2_ host material, as no obvious lattice fringes can be observed in the HRTEM images (Figure 1b,c). This can be further supported by the selected electron diffractions shown in Figure 1e, where only a diffuse halo can be found in the image [33]. After careful observations, it is interesting to find that there are some dark spots with diameter sizes ranging from 1 nm to 3 nm distributed on the surface of UiO-66-NH_2_. These dark spots are assigned to the BCQDs, demonstrating the successful functionalization of UiO-66-NH_2_ with BCQDs. In addition, the tight contact interface shown in the HRTEM images is indicative of the strong interactions between the UiO-66-NH_2_ host and BCQDs guest, originating from the mutual interactions between the surface groups (hydroxyl and amino group) of these two materials. As the C, N, O, S, and P elements are derived from the BCQDs according to our previously reported results [9,32], the observation of well-distributed C, N, O, S, and P on the entire MOFs shown in the EDX mappings is indicative of a uniform distribution of BCQDs on the UiO-66-NH_2_. Moreover, it is interesting to note the uniform distribution of Cl^−^ on the entire material. Since the Cl is not a constituent element of both UiO-66-NH_2_ and BCQDs, the observation of Cl distributing well on the materials is indicative of the successful protonation of BCQDs/UiO-66-NH_2_ that introduces Cl as charge compensation to that of H^+^. These results demonstrate the successful construction of H^+^/BCQDs/UiO-66-NH_2_. 

In energy- and environmental-related areas, the difficulty in catalyst recycling is a bottleneck that restricts the practical application of powder-form catalyst. To overcome this drawback, we then chose ceramsite as a host material to immobilize the powder-form catalyst of H^+^/BCQDs/UiO-66-NH_2_, forming an easily recyclable immobilized catalyst of H^+^/BCQDs/UiO-66-NH_2_@ceramsite. As depicted in Figure 2a, the ceramsite features a brown color with a round shape, while the color changed dark brown after the loading of H^+^/BCQDs/UiO-66-NH_2_. The dark color shown here not only suggests the successful immobilization of catalyst on the ceramsite, but also reflects the excellent light absorption property of the as-prepared immobilized catalyst [22,34] (which can be further confirmed by the DRS results shown in Figure 5c). Furthermore, it is easy to find in Figure 2a that there are abundant pores on the surface of the ceramsite, which may help to immobilize the catalyst. This can be further supported by the elemental mappings shown in Figure 2, where the H^+^/BCQDs/UiO-66-NH_2_ composed elements are found to be well distributed on the ceramsite. Especially, the Cl element is also found distributed on the material with an atomic content of 1.7%, demonstrating the successful installation of active sites on the immobilized material. Based on the above analysis, the immobilized H^+^/BCQDs/UiO-66-NH_2_@ceramsite catalyst has been successfully fabricated.

It is believed that the catalytic activity of a catalyst is sensitive to the porosity structure, which affects the mass transfer and exposure number of active sites; thus, the as-prepared materials of UiO-66-NH_2_, H^+^/UiO-66-NH_2_, BCQDs/UiO-66-NH_2_, and H^+^/BCQDs/UiO-66-NH_2_ were further investigated by performing a nitrogen sorption–desorption isothermals test. As shown in Figure 3a, the surface areas of UiO-66-NH_2_, H^+^/UiO-66-NH_2_, BCQDs/UiO-66-NH_2_, and H^+^/BCQDs/UiO-66-NH_2_ were respectively tested and are shown to be 511, 535, 311, and 661 m^2^·g^−1^. It should be noted here that the acidification process could affect the porosity structure via H^+^ corrosion, thus causing H^+^/UiO-66-NH_2_ and H^+^/BCQDs/UiO-66-NH_2_ to exhibit a much higher surface area than that of the UiO-66-NH_2_ and BCQDs/UiO-66-NH_2_ counterparts. It is worth noticing here that the introduction of BCQDs on the MOFs led to a decrement in surface area by blocking the pores of the MOFs, as supported by the surface area decrement from 511 to 311 m^2^·g^−1^. However, the H^+^ etching of both the BCQDs and UiO-66-NH_2_ created abundant pores on both components. This can be further evidenced by the pore size distribution results shown in Figure 3b, in which the H^+^/BCQDs/UiO-66-NH_2_ shows much higher pore volume that that of BCQDs/UiO-66-NH_2_. In addition, no obvious structure destruction can be found from the FT-IR results in Appendix A, as all the materials show comparable FT-IR spectra, evidencing that the acidification treatment does not affect the skeleton of the MOFs. Furthermore, all the materials exhibited high porosity with pore sizes ranging from several nanometers to several tens of nanometers, which may have had a positive effect on the catalytic reaction by promoting the mass transfer as well as light absorption via the multi scattering and reflection of the incident light. 

The chemical structure of as-prepared catalysts of UiO-66-NH_2_, H^+^/UiO-66-NH_2_, BCQDs/UiO-66-NH_2_, and H^+^/BCQDs/UiO-66-NH_2_ were further investigated by XPS. As shown in Figure 4, the representative elements of C, N, O, S, P, Zr, and Cl can be clearly observed in the XPS survey and high-resolution spectra, in accordance with the EDX mapping results shown in Figure 1, which again confirms the successful construction of H^+^/BCQDs/UiO-66-NH_2_ catalyst. Furthermore, all the materials produced comparable C 1s, N 1s, O 1s, and Zr 3d XPS spectra, demonstrating the high structural stability of UiO-66-NH_2_ against the modification of BCQDs and acidification treatment [3,5]. It should be noted here that the abundant oxygen-containing and amino groups on the surface of the materials would allow the strong interactions between BCQDs and UiO-66-NH_2_. This is further supported by the observation of the shift of O 1s and Zr 3d XPS spectra of BCQDs/UiO-66-NH_2_ toward a higher binding energy direction than that of the UiO-66-NH_2_ counterpart [3,5,6]. Surprisingly, the O 1s and Zr 3d XPS spectra of H ^+^/BCQDs/UiO-66-NH_2_ give an additional shift toward higher binding energy compared with that of BCQDs/UiO-66-NH_2_, suggesting that the acidification treatment could have further promoted the interactions between BCQDs and UiO-66-NH_2_. 

As the acidification of as-prepared material of BCQDs/UiO-66-NH_2_ facilitated the interactions between BCQDs and UiO-66-NH_2_ and showed a shift of XPS spectra toward the blue direction, we then used this phenomenon as an indicator to probe the successful introduction of protons on the BCQDs/UiO-66-NH_2_. As shown in Figure 4e, the Zr 3d spectrum of BCQDs/UiO-66-NH_2_ is deconvoluted into two peaks with binding energies of 182.5 and 184.9 eV, respectively, assigned the Zr 3d_5/2_ and 3d_3/2_ levels [35], while the peaks located at 168.34 and 169.38 eV refer to the S 2p_3/2_ and 2p_1/2_ levels [36]. Moreover, the two peaks with binding energies of 133.53 and 134.64 eV are attributed to the P 2p levels [37]. The more exciting thing is that the Zr 3d, S 2p, as well as P 2p spectra of BCQDs/UiO-66-NH_2_ shift toward the higher binding energy direction, indicating the successful introduction of proton sites on the composites after the facile hydrochloride acid treatment. This can be further confirmed by the Cl 2p spectra in Figure 4h, in which a clear Cl 2p signal is found on the material after the acidification treatment. The introduction amount of proton is about 1.5 atomic% of H^+^/BCQDs/UiO-66-NH_2_, which is deduced from the atomic amount of Cl in the composite. 

### 3.2. Catalytic Performance of the Powder-Form Catalysts

We first evaluated the catalytic performance of as-prepared powder-form catalysts by using an electric furnace to heat the reaction flask so as to hold the reaction temperature of 100 °C. In this work, a combined BCQDs modification and acidification treatment was employed to functionalize the UiO-66-NH_2_ host. The amount of BCQDs introduced and the concentration of hydrochloric acid used for the acidification are two key factors determining the catalytic performance of H ^+^/BCQDs/UiO-66-NH_2_. In this regard, a systematic study was carried out, and the results are shown in Figure 5a,b. As shown in Figure 5a, the EMF yield increased from 46% to 55% with the increase in the BCQDs loading amount, which is attributed to the increase in catalytic sites. However, when the BCQDs concentration used for H^+^/BCQDs/UiO-66-NH_2_ synthesis was too high, the excess BCQDs aggregated into large carbon, which not only led to the sharp decrement of active sites for the catalytic reaction, but more importantly decreased the EMF production selectivity. This can be further supported by the selectivity decrement from 67% to 24% shown in Figure 5a.

To investigate the effect of the HCl concentration used for the catalyst acidification on the catalytic activity toward the conversion of HMF to EMF, HCl was diluted 1, 2, 4, and 8 times before the acidification treatment. As depicted in Figure 5b, increasing the HCl concentration gave an increment in the selective production of EMF, due to the fact that the acidification of a catalyst in a low-HCl-concentration solution cannot fully guarantee the introduction of adequate H^+^ sites. The maximum EMF production activity was achieved when the HCl used for catalyst acidification was diluted 2 times, where the EMF yield reached 55% with the HMF-to-EMF selectivity of 66.9%. However, without diluting the HCl solution, the concentrated HCl with a highly corrosive property would largely etch the catalyst, leading to structure damage. As a result, both the EMF production yield and the HMF-to-EMF selectivity decreased from 55% and 66.9% to the final 28.9% and 65.1%, respectively. Based on the above results, the optimized conditions for catalyst synthesis were established and the optimized condition was used for the materials construction thereafter. 

We further investigated the catalytic performance of UiO-66-NH_2_, H^+^/UiO-66-NH_2_, BCQDs/UiO-66-NH_2,_ and H^+^/BCQDs/UiO-66-NH_2_, where the materials were fabricated based on the aforementioned optimized synthetic method. As shown in Figure 5c, no obvious EMF yield could be found, neither for UiO-66-NH_2_ nor for BCQDs/UiO-66-NH_2_, highlighting the importance of introduced H^+^ sites for converting HMF into EMF. It is interesting to find that although the EMF yield on these two catalysts was zero, more than 45% of HMF was converted. The HFM conversion on these two materials was due to the side reactions that covert HMF into other kinds of byproducts. After the acidification treatment, a noticeable EMF yield was then found on H^+^/UiO-66-NH_2_; however, the selectivity for the etherification of HMF into EMF was as low as 4.7%; while the selectivity for the etherification of HMF to EMF was almost 100%, demonstrating that the introduction an of H^+^ site on UiO-66-NH_2_ cannot prevent the unwanted serious side reaction. Excitingly, H^+^/BCQDs/UiO-66-NH_2_ exhibited excellent catalytic activity toward EMF generation, where the EMF yield and selectivity values reached as high as 55% and 66.9%, respectively. The enhanced catalytic activity should lead to the strong interactions between the BCQDs guest and UiO-66-NH_2_ host that suppresse the side reactions during the etherification of HME to EMF [38], which is agreement with the XPS results shown in Figure 4.

Before performing the IR-light-triggered etherification reactions, we first tested the light absorption property of the as-prepared catalysts, as the light absorption ability is a key factor affecting the IR light driving catalytic performance. As shown in Figure 5d, all the powder-form materials show two typical absorption regions, i.e., UV-Vis and IR regions. However, the materials with the modification of BCQDs show excellent near-UV-Vis and near-IR light absorption ability, assigned to the black-colored BCQDs possessing high UV-Vis and IR light absorption. We then immobilized the powder-form catalyst of H^+^/BCQDs/UiO-66-NH_2_ on to the ceramsite so as to make it easy for catalyst recycling. As shown in Figure 5d, the H^+^/BCQDs/UiO-66-NH_2_@ceramsite shows excellent light absorption ability in the full range of solar light. Such an excellent light absorption, especially for the high IR light absorption property shown here, is beneficial for photo-to-thermal conversion. 

### 3.3. Photo-to-Thermal Conversion and Catalytic Performance of the Immobilized Catalysts

With a high light absorption property, an infrared thermal imaging camera was employed to evaluate the photo-to-thermal conversion performance of H^+^/BCQDs/UiO-66-NH_2_@ceramsite. Here, a 150 W IR light was the IR source used to simulate the solar light. As shown in Figure 6, without IR light irradiation, the temperature of the reaction system was about 24 °C, comparable to the room temperature. As expected, the temperature of the ceramsites rose rapidly from room temperature to about 68 °C after only one hour IR light irradiation. The temperature further increased with an increase in irradiation time, where the temperature of ceramsite reached 95 °C and 138 °C after 2 and 3 h irradiation, respectively. However, the temperature of the reaction system would not further rise regardless of the prolonged irradiation times and remained at around 138 °C, which was due to the solvent evaporation under a such a high temperature. It should be noted here that the temperature originating from the photo-to-thermal conversion was high enough (higher than 100 °C) to drive the etherification of HMF with ethanol into EMF. 

Based on the above analysis, we then carried out the catalytic etherification reaction by using the immobilized material of H^+^/BCQDs/UiO-66-NH_2_@ceramsite using IR as input driven energy. Firstly, we studied the effect of calcination temperature during the immobilization of BCQDs/UiO-66-NH_2_ on the ceramsite on the final catalytic activity. As shown in Figure 7a, the EMF yield increased from 44% to 73% with the increase in annealing temperature from 250 to 300 °C. The enhanced catalytic activity was due to the strengthened interactions between the BCQDs and UiO-66-NH_2_, which suppressed the side reaction during the etherification of HMF to EMF. However, further raising the annealing temperature would more or less destroy the skeleton of the MOFs, thus reducing the catalytic activity. In addition, the higher temperature would inevitably lead to the sintering of the material, effecting the interactions between the BCQDs and UiO-66-NH_2_ [39]. As a result, both the EMF yield and HMF-to-EMF selectivity were decreased. Based on the above analysis, we can conclude here that the optimized post calcination temperature to strengthen the interactions between BCQDs and UiO-66-NH_2_ as well as the adhesive of BCQDs/UiO-66-NH_2_ on ceramsite is 300 °C.

To highlight the merits of using IR light instead of electric heat as driven energy to open the etherification reaction, we then evaluated the catalytic activity of H^+^/BCQDs/UiO-66-NH_2_@ceramsite under these two conditions. As shown in Figure 7b, as the powder-form catalyst dispersing in the solution inevitably sunk to the bottom of the reactor, the IR could not transmit through the solvent to reach the sinking catalyst, and as a result, the surface temperature of the catalyst was rather low and could not drive the etherification reaction. This could explain why the catalytic performance of powder-form catalyst was much lower than that of the catalyst heated by electric furnace. After immobilization, the catalyst on the lightweight ceramsite was floating over the solution; thus, most of the IR light could reach the catalyst and convert into thermal energy, which in return guaranteed the temperature of reaction system above the etherification temperature. Surprisingly, the immobilized catalyst triggered by IR irradiation exhibited comparable catalytic activity toward the conversion of HMF to EMF to the one triggered by electric furnace. The EMF yield and selectivity over the IR-triggered system respectively reached 45% and 65%. Furthermore, an additional catalytic activity improvement was observed on the immobilized catalyst compared to the powder-form catalyst when using an electric furnace as input driven energy, originating from the strengthened interaction between BCQDs and UiO-66-NH_2_ by high-temperature calcination during the immobilization process. 

It is well accepted that catalyst stability is a very importance factor determining the large-scale practical application of the immobilized catalyst. In this regard, the stability of H^+^/BCQDs/UiO-66-NH_2_@ceramsite was evaluated by four consecutive catalytic etherification cycles of HMF with ethanol to EMF. After each cycle, the immobilized catalyst was taken out of the system and washed with deionized water at least three time, and the catalyst was further dried before using it in the next cycle. As shown in Figure 7c, however, the catalytic performance decreased sharply from the initial 45% to 18% of EMF yield for the second round. The EMF yield further decreased to less than 10% after the third round, and such a serious catalytic activity loss would restrict the catalyst reuse. It is necessary to uncover what makes the catalyst deactivate, as catalyst stability is a key determining the practical application. In catalysis, the organic species adsorbing on the catalyst is a common phenomenon poisoning the catalytic activity [40]; thus, we deduce that this may also be the very reason leading to the catalytic activity loss. Generally, the absorbed organic species cannot be desorbed merely by water washing; thus, we then used acid to wash the catalyst so as to facilitate the desorption of these species off the catalyst. As expected, the catalytic activity recovered to the initial one, where the EFM yield (44.5%) is comparable to the one used for the first run. Therefore, to avoid the loss of catalytic activity, the catalyst should be washed with acid solution for poisoning species desorption so as to maintain the high catalytic activity. Based on the above analysis, the corresponding synthetic routes for the fabrication of H^+^/UiO-66-NH_2_, H^+^/BCQDs/UiO-66-NH_2_ and H^+^/BCQDs/UiO-66-NH_2_@ceramsite are then illustrated in Figure 1. Furthermore, the mechanistic insight for the etherification of HMF with ethanol into EMF on various catalysts is given in Figure 1.

## 4. Conclusions

In this study, powder and immobilized heterogeneous acid catalysts H^+^ /BCQDs/UiO-66-NH2 and H^+^ /BCQDs/UiO-66-NH2@Ceramsite were successfully prepared through simple steps using low-cost raw materials. The as-prepared catalysts showed considerably high catalytic activity toward the synthesis of EMF from HMF owing to the large specific surface area of UiO-66-NH2; a large number of acid catalytic active sites were exposed on the catalyst surface. The BCQDs and H^+^ can be successfully loaded onto the surface of UiO-66-NH2 due to the surface −NH2 groups on the UiO-66-NH2. Based on the catalytic test results, the highest acid catalytic activity of H^+^ /BCQDs/UiO-66-NH2 toward the directional transformation of HMF among all the tested materials can be attributed to the internal synergy between BCQDs and H^+^ in H^+^ /BCQDs/UiO-66-NH2. Moreover, the prepared immobilized acid catalyst H^+^ /BCQDs/UiO-66-NH2@Ceramsite showed pretty infrared absorption characteristics. The heat collection property of ceramsite carrier causes the H^+^ /BCQDs/UiO-66-NH2@Ceramsite to exhibit excellent acid catalytic activity regardless of heating or infrared irradiation. Therefore, the prepared H^+^ /BCQDs/UiO-66-NH2@Ceramsite catalyst is not only expected to be applied to industrial-scale production, but is also a broad prospect for photothermal co-catalysis.

## Data Availability

Not applicable.

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
