# Peer review of "Carbon Quantum Dots-Functionalized UiO-66-NH2 Enabling Efficient Infrared Light Conversion of 5-Hydroxymethylfurfuryl with Waste Ethanol into 5-Ethoxymethylfurfural"

_ijerph, 2022, doi:10.3390/ijerph191610437_

Round 1
Reviewer 1 Report
In this study, This study demonstrates a clever design and construction of an immobilized catalyst H+/bcqds/uio-66-NH2@ceramsite for IR light triggered etherification of HMF with waste ethanol into EMF. The modified bcqds not only sever as a photo-to-thermal conversion carrier to guarantee the high temperature for the etherification reaction. The as-prepared H+/bcqds/uio-66-NH2@ceramsite exhibits excellent IR light triggered catalytic activity toward the conversion Of HMF into EMF. The EMF yields and selectivity are tested as high as 45% and 65%, respectively. The innovative design opens a new avenue in designing IR-triggered catalysts for high-efficient bio-fuels synthesis. Experiments were conducted roundly and reasonably and the results were exhibited and discussed rigorously and scientifically. Therefore, I’d like to recommend this manuscript for publication after resolving the minor issues noted below.
1. Please add more result data in the Abstract
2. The author should add more previous literature in the Introduction section and improve the whole section
3. Section 2.2 Material preparation, author should also write chemical compound in a mmol unit
4. Figure 5c, no obvious EMF yield can be found neither for UiO-66-NH2 nor BCQDs/UiO-66-NH2, highlighting the importance of introduced H+ site for converting HMF into EMF Why? So without H+ site, CQDs/UiO-66-NH2 material is not efficient?
5. Page No 8. In that paragraph, please give the previous references to verify this Data
6. The author should improve the Conclusion
7. The author should be careful about author guidelines regarding whole manuscript and also the format of the reference, please recheck.
Reviewer 2 Report
The research on the carbon quantum dots functionalized UiO-66-NH2 Eenabling efficient infrared light conversion of 5-hydroxymethylfurfuryl with waste ethanol into 5-ethoxymethylfurfural is shown in this paper.
In the era of rapid development of biofuels, the subject of obtaining a catalyst for high-efficiency biofuel synthesis is very important. For this reason, research on this type of catalysts seems even necessary.
Indeed the conceptual study shown here provides new viewpoints in designing energy effective materials for the conversion of wastes into high value added resources.
References have been well selected the content of the paper.
Comments:
- It should be good to rewrite the introduction and the conclusions to for better understanding of the text.
